# TS-DDAE: A Novel Temporal-Spectral Denoising Diffusion Autoencoder for Wireless Signal Recognition Model Pre-training

**Yaoqi Liu**[1,2]**, Jin Wang**[2]**, Hui Wang**[2]**, Chuan Shi**[1,2*]
Beijing University of Posts and Telecommunications[1]
Peng Cheng Laboratory[2]
`yaoqiliu@bupt.edu.cn`, `{wangj05,wangh06}@pcl.ac.cn`,
`shichuan@bupt.edu.cn`

## Abstract

Wireless Signal Recognition (WSR) aims to identify the property of received signals using Artificial Intelligence (AI) without any prior knowledge, which has been widely used in civil and military radios. The current AI trend of pre-training and fine-tuning has shown great performance, and the existing pre-trained WSR models also achieve impressive results. However, they either apply the "mask-reconstruction" pre-training strategy, which may disrupt intricate local dependencies of signals, or overlook latent spectral characteristics. Therefore, in this paper, we follow the diffusion models and propose a pre-training framework for WSR, named the Temporal-Spectral Denoising Diffusion AutoEncoder (TS-DDAE), which learns signal representations by corrupting signals with temporal and spectral noise, and then reconstructing the original data with a learned neural network. Moreover, we design a novel neural architecture, named TS-Net, which couples self-attention for temporal encoder with channel attention for spectral encoder. Extensive experiments on several datasets and WSR tasks show that TS-DDAE achieves superior performance compared to state-of-the-art (SOTA) baselines, which demonstrate the potential to be a foundation model for WSR. Code is available at `https://github.com/BUPT-GAMMA/FoundWSR`.

## 1 Introduction

Various signals are generated and play key roles in our daily communication activities Vinciarelli et al. (2008). As the scale of signal data continues to expand, intelligent communication technology Huang et al. (2018) that uses artificial intelligence (AI) Wang et al. (2020) for signal analysis is gradually becoming one of the most important characteristics of the sixth-generation (6G) network due to its high efficiency and accuracy Alsharif et al. (2020). Wireless signal recognition (WSR) Li et al. (2019) is to identify the basic property without any prior knowledge of the signals, such as its modulation type Meng et al. (2018) and wireless technology Bitar et al. (2017), which requires experts to capture subtle features in the time and the spectral domains. Once prior knowledge of the signals is known, the receiver can choose the corresponding demodulator, and later processing steps run faster and more reliably Eldemerdash et al. (2016). WSR is now routine in civil and military radios and is a basic building block of intelligent communication systems Li et al. (2019).

Currently, deep learning models for WSR such as IQFormer Shao et al. (2024) have delivered competitive accuracy on individual benchmark, while they are difficult to generalize to multiple tasks. The pre-training and fine-tuning paradigm Han et al. (2021) has proven that generic representations can be fine-tuned to a wide range of downstream objectives with minimal effort. However, wireless signals have not yet enjoyed the same benefit. The few existing models like SpectrumFM Zhou et al. (2025) choose the "mask-reconstruction" strategy Devlin et al. (2019) by setting the amplitude of part of the signal to 0 and restoring the original one, which risks scrambling the delicate temporal-spectral structure of waveforms Yang et al. (2023). Moreover, these methods usually fo-

---

*Corresponding author

cus exclusively on time-series, ignoring the rich information embedded in the spectral domain. A robust pre-training framework that respects the inherent temporal and spectral nature of wireless signals remains an open challenge.

To retain the original data information rather than directly eliminating it, we refer to the "noise-reconstruction" strategy proposed by diffusion models Ho et al. (2020), which is adding random Gaussian noise to the clean data and restoring it as the pre-training objective, and propose our framework, named the **T**emporal **S**pectral **D**enoising **D**iffusion **A**uto**E**ncoder (TS-DDAE) for WSR pre-training. The pre-training strategy used by the TS-DDAE can maintain the temporal-spectral structure of the original waveform compared to the "mask-reconstruction" strategy, and the TS-DDAE applies a neural network to reconstruct the signal from both temporal and spectral perspectives. Following the diffusion paradigm, the TS-DDAE contains a forward Markov process that incrementally corrupts the data and a learned backward process that iteratively restores it. During the forward phase, we add random noise to both the time and the spectral domains to form noisy signals. In the backward phase, we design a novel neural network architecture named TS-Net to reconstruct the original data. The TS-Net can be divided into the temporal encoder and the spectral encoder, where we apply the self-attention Hu (2019) for the temporal encoder and the channel attention Guo et al. (2022) for the spectral encoder. The encoders interact with each other to jointly learn the temporal and spectral characteristics of the signal data. Pre-trained and fine-tuned on various signal datasets and WSR tasks have achieved state-of-the-art (SOTA) performance, with 1.32% improvements on average over the best baseline, and about 8.75% improvements on average compared to IQFormer Shao et al. (2024), the SOTA Automatic Modulation Classification (AMC) model. Supplementary and ablation experiments also demonstrate the effectiveness of our designed model.

We summarize our contributions as follows.

- To the best of our knowledge, we are the first to adapt the diffusion theory to pre-train models for WSR. We formulate the TS-DDAE, which contains a principled self-supervised learning objective from both temporal and spectral perspectives.

- We design a novel neural network architecture named TS-Net that jointly refines temporal sequence and spectral structure, allowing each encoder to inform the other and extract a richer, complementary representation of signals.

- Related experiments demonstrate the effectiveness of our solution with most of the SOTA performance compared to 11 baseline models. Furthermore, we provide a code repository in supplementary materials that integrates signal data processing and various WSR models to facilitate user-replicated benchmarks and the design of their own solutions.

## 2 RELATED WORK

### 2.1 DENOISING DIFFUSION MODELS

In the past five years, Denoising Diffusion Models (DDM) Yang et al. (2023) have rapidly become a cornerstone of modern generative modeling and have demonstrated unprecedented success in computer-vision (CV) tasks such as high-definition image generation Dong et al. (2021), text-to-image generation Li et al. (2023), etc. The most typical DDPM Ho et al. (2020) has established the "noise-and-denoise" paradigm, which is to add noise to the data in the forward process, and restore it step by step in the backward process to train the generative abilities of a neural network such as U-Net Ronneberger et al. (2015). The DDIM Song et al. (2021) further expands the solution space based on the DDPM and sets the variance as a hyperparameter. Beyond image generation, recent studies show that DDM can also serve as a powerful self-supervised pre-training objective. The DDAE Xiang et al. (2023) takes the "noise-and-denoise" paradigm as its pre-training task, which has also achieved a superior image classification Sanghvi et al. (2020) performance.

In the area of signals, RF-Diffusion Chi et al. (2024) has also adapted the diffusion theory to conditional signal generation. However, our goal is not to generate signals but to learn robust representations for WSR without any conditional assumptions. Therefore, we depart from RF-Diffusion and instead build upon the theory of the DDAE and propose an unconditional diffusion-based pre-training pipeline tailored to wireless signals.

## 2.2 WIRELESS SIGNAL RECOGNITION MODELS

Traditional signal analysis methods usually rely on hand-crafted statistics for tasks such as AMC, but crafting these features is labor intensive Dobre et al. (2007). Therefore, CNN and Transformer Li et al. (2021) Vaswani et al. (2017) based deep learning models have emerged to replace most manual feature extraction, such as AMC_Net Zhang et al. (2023), MSNet Zhang et al. (2021), etc. Recent studies, such as IQFormer Shao et al. (2024), jointly model raw signal samples and their spectrograms, achieving excellent performance in AMC task.

Self-supervised pre-training strategies for wireless signals can be grouped into five categories: deep clustering methods, contrastive methods, reconstruction-based methods, generative methods, and predictive methods Milosheski et al. (2025). Among them, SpectrumFM Zhou et al. (2025) borrows the idea of masked-language-modeling from NLP, randomly masks signals, and reconstructs them to build a foundation model that supports multiple WSR tasks. In contrast, we corrupt the original signals by adding noise to both the time and the spectral domains, and train the neural network to denoise the corrupted signals. As corruption is additive rather than masking, the input does not lose too much content, so the pre-text task retains finer-grained information while still requiring the model to learn semantically rich, transferable representations.

# 3 METHODOLOGY

## 3.1 OVERVIEW

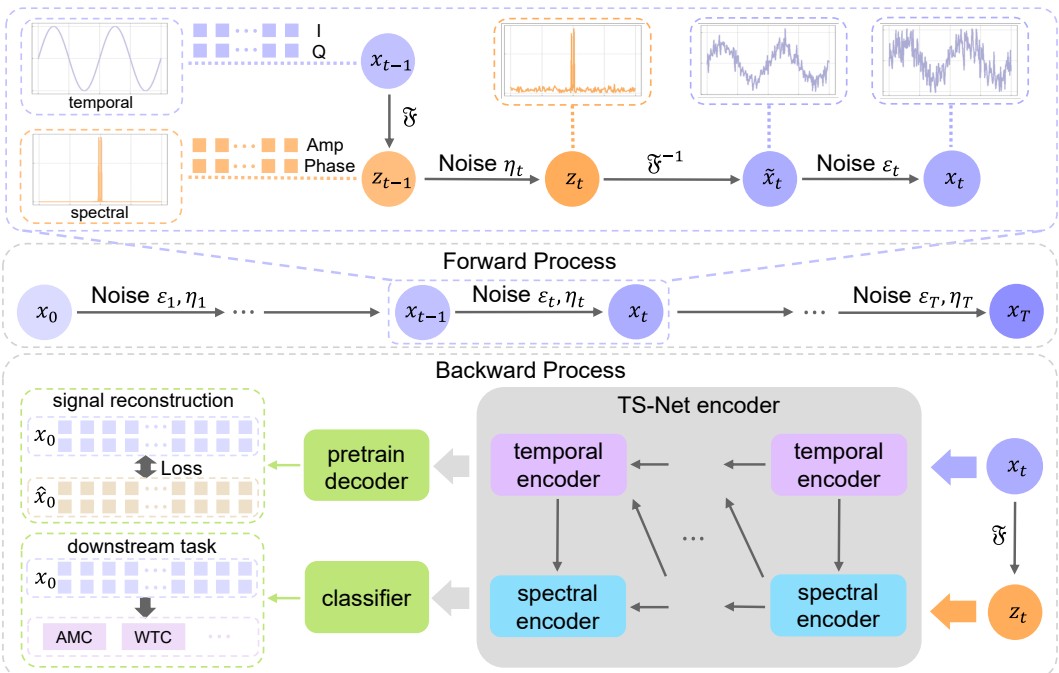

Figure 1: The overview of the TS-DDAE framework.

The real-world received signal is usually represented as IQ data, where I and Q are the in-phase and quadrature components, respectively, which is often represented using complex data, i.e. $x = I + jQ, j = \sqrt{(-1)}$. However, we believe that the two components are related and should be considered simultaneously when analyzing signals. Moreover, the real data calculations are usually more efficient than the complex data. Therefore, we represent the IQ data of length $L$ as a two-row matrix $x \in \mathbb{R}^{2 \times L}$, where the two rows represent the I and Q components, respectively. However, users can only obtain limited information through time series. Most signal analysis methods convert IQ data into spectral data by applying the Fourier transformation $z = \mathfrak{F}(x) \in \mathbb{R}^{2 \times L}$, which describes

the distribution of the signal amplitude across frequencies. The spectrum of signal data can better reflect the essential characteristics of the signal.

Both the time series and spectrum of IQ data contain strong local dependencies, so pre-train a WSR model using "mask-reconstruction" will seriously damage the original information. Therefore, we take the "noise-reconstruction" strategy from the diffusion models and propose a novel diffusion theory named Temporal-Spectral Denoising Diffusion AutoEncoder (TS-DDAE), which is shown in Figure 1. During the forward process, we corrupt both the time and the spectral domains of the IQ data by adding the corresponding Gaussian noise, which can perturb the data while preserving its original characteristics. To address the two potential distortions, the reverse learning process needs to restore not only the temporal characteristics of the original data, but also the spectral characteristics. Therefore, we specially design a hybrid **T**emporal-**S**pectral neural **Net**work (TS-Net), which uses temporal encoders and spectral encoders to jointly learn the feature representations of the signal in the time and the spectral domains. At the pre-training stage, the TS-DDAE learns to reconstruct the original IQ data from the noisy one. At the fine-tuning stage, we optimize the model with task-specific data to adapt to downstream tasks like AMC, WTC, etc. Detailed TS-DDAE and TS-Net are introduced in Section 3.2 and Section 3.3, respectively.

## 3.2 TEMPORAL-SPECTRAL DENOISING DIFFUSION MODEL

**Forward Process**. Given IQ signal data $x_0 \in \mathbb{R}^{2 \times L}$, the forward process gradually corrupts the temporal and spectral information of signals by adding $T$ steps of Gaussian noise. The results of each step can be denoted as $x_1, x_2, \cdots, x_T$. To corrupt the spectrum, we need to apply the Fourier transformation $z_0 = \mathfrak{F}(x_0)$, and formulate the spectral noise addition process as

$$z_t = \tilde{\mu}_t \mathfrak{F}(x_{t-1}) + (\tilde{\tau}_t/\sigma) \cdot \delta_t, \tag{1}$$

where $\tilde{\mu}_t^2 + (\tilde{\tau}_t/\sigma)^2 = 1$, and $\delta_t \sim \mathcal{N}(0, \sigma)$. Then, we add noise to the time domain,

$$x_t = \tilde{\alpha}_t \mathfrak{F}^{-1}(z_t) + \tilde{\beta}_t \varepsilon_t, \tag{2}$$

where $\tilde{\alpha}_t^2 + \tilde{\beta}_t^2 = 1$, and $\varepsilon_t \sim \mathcal{N}(0, I)$, which is a standard Gaussian noise. By combining Equation 1 and 2, we can formulate the noise addition from step $t-1$ to $t$ as

$$x_t = \tilde{\alpha}_t \mathfrak{F}^{-1}(\tilde{\mu}_t \mathfrak{F}(x_{t-1}) + (\tilde{\tau}_t/\sigma) \cdot \delta_t) + \tilde{\beta}_t \varepsilon_t = \tilde{\alpha}_t \tilde{\mu}_t x_{t-1} + \tilde{\alpha}_t \cdot (\tilde{\tau}_t/\sigma) \cdot \mathfrak{F}^{-1}(\delta_t) + \tilde{\beta}_t \varepsilon_t. \tag{3}$$

The inverse Fourier transformation of a Gaussian function is still a Gaussian function. Therefore, we denote $\eta_t = \mathfrak{F}^{-1}(\delta_t) \sim \mathcal{N}(0, I)$, which we require to be a standard Gaussian noise. Furthermore, we denote $\alpha_t = \tilde{\alpha}_t \tilde{\mu}_t, \gamma_t = \tilde{\alpha}_t \frac{\tilde{\tau}_t}{\sigma}$, and simplify $\tilde{\beta}_t$ as $\beta_t$. The Equation 3 can be rewritten as

$$x_t = \alpha_t x_{t-1} + \beta_t \varepsilon_t + \gamma_t \eta_t, \tag{4}$$

where the coefficients satisfy $\alpha_t^2 + \beta_t^2 + \gamma_t^2 = 1$. By iterating the Equation 4 recursively and incorporating the reparameterization trick Kingma et al. (2015), we can describe the relationship between the IQ data $x_0$ and noisy data $x_t$ as

$$x_t = \bar{\alpha}_t x_0 + \bar{\beta}_t \bar{\varepsilon}_t + \bar{\gamma}_t \bar{\eta}_t, \tag{5}$$

where $\bar{\alpha}_t = \prod_{i=1}^{t} \alpha_i, \bar{\beta}_t = \sum_{i=1}^{t} (\frac{\bar{\alpha}_t}{\bar{\alpha}_i} \beta_i), \bar{\gamma}_t = \sum_{i=1}^{t} (\frac{\bar{\alpha}_t}{\bar{\alpha}_i} \gamma_i), \bar{\varepsilon}_t, \bar{\eta}_t \sim \mathcal{N}(0, I)$, and $\bar{\alpha}_t^2 + \bar{\beta}_t^2 + \bar{\gamma}_t^2 = 1$. Therefore, we can directly obtain the noisy data from the original IQ data given any step $t$.

**Backward Process**. After we get the noisy data, we try to restore it back to the original IQ data. Here, we describe this process using the Bayesian theorem. According to the superposition of the Gaussian distribution, Equation 5 can be rewritten as a Gaussian distribution conditioned on $x_0$,

$$p(x_t|x_0) \sim \mathcal{N}(x_t; \bar{\alpha}_t x_0, (\bar{\beta}_t^2 + \bar{\gamma}_t^2)I). \tag{6}$$

Since the added noise from each step does not affect each other during the forward process, it can be approximated as a Markov process, i.e. $p(x_t|x_{t-1}, x_0) \approx p(x_t|x_{t-1}) \sim \mathcal{N}(x_t; \alpha_t x_{t-1}, (\beta_t^2 +$

$\gamma_t^2)I)$. Therefore, we can get the probability function $p(x_{t-1}|x_t, x_0)$ for the backward process with the Bayes' theorem. Detailed formulation is shown in Appendix A. However, $p(x_{t-1}|x_t, x_0)$ is conditioned on $x_0$, which is unseen during the backward process. We can apply a neural network (NN) to fit $x_0$. Suppose that we use the NN $\bar{\mu}(x_t)$ to estimate $x_0$ based on $x_t$, the loss function can be defined as $||x_0 - \bar{\mu}(x_t)||$. Following the Equation 5, the NN can be formulated as $\bar{\mu}(x_t) = \frac{1}{\bar{\alpha}_t}(x_t - \bar{\beta}_t \varepsilon_{\theta_1}(x_t, t) - \bar{\gamma}_t \eta_{\theta_2}(x_t, t))$. Then, our final loss function is

$$\mathcal{L}(x_t, t) = \frac{\bar{\beta}_t^2}{\bar{\alpha}_t^2}[(\varepsilon - \varepsilon_{\theta_1}(x_t, t)) + \lambda(\eta - \eta_{\theta_2}(x_t, t))]^2, \varepsilon, \eta \sim \mathcal{N}(0, I), \quad (7)$$

where $\lambda = \bar{\gamma}_t / \bar{\beta}_t$ can be seen as a hyperparameter, which is the ratio between the spectral noise and temporal noise intensity in the noisy data, and $\theta_1, \theta_2$ are learnable parameters. During pre-training, we need to sample IQ data $x_0$ from the dataset, a diffusion step $t$, and two noise $\varepsilon, \eta$ from a standard Gaussian distribution.

Although, formally, Fast Fourier Transformation (FFT) is a unitary transform, the two noises are sampled separately from the Gaussian distribution and are not completely identical, thus forming two optimization objectives for joint optimization. Future work could explore alternative noise addition schemes better suited to the characteristics of wireless signals.

## 3.3 MODEL ARCHITECTURE

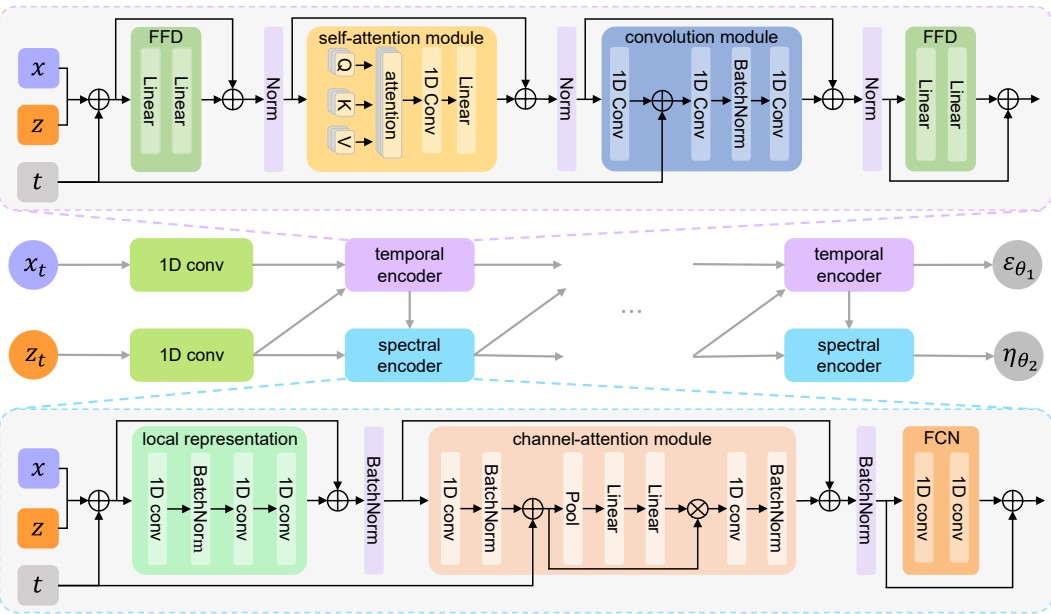

Figure 2: The model architecture of the TS-Net encoder.

To jointly capture temporal structure and spectral signatures, we introduce TS-Net, a diffusion-inspired architecture that performs hybrid analysis in both domains. The detailed TS-Net architecture is shown in Figure 2. The TS-Net comprises two tightly coupled encoders based on Equation 7. The temporal encoder learns to denoise raw IQ samples $x_t$ by predicting the temporal noise $\varepsilon_{\theta_1}$, while the spectral encoder operates on the spectral counterpart $z_t = \mathfrak{F}(x_t)$ to estimate the spectral noise $\eta_{\theta_2}$. Before feeding into each encoder, a 1D convolution layer (denoted as Conv1D) with kernel size $k = 1$ projects the respective input into a $C$-dimensional embedding space, simulating the initial processing of complex-valued IQ data.

$$\boldsymbol{X}^{conv} = \text{Conv1D}(\boldsymbol{X}), \boldsymbol{X} \in \mathbb{R}^{B \times 2 \times L}, \quad (8)$$

where $B$ is the batch size of the data, and the default kernel size of Conv1D is 1. At the same time, the diffusion step $t$ is also encoded with sinusoidal position embeddings from Transformer.

Then, following the theory outlined in Section 3.2, we input the IQ sequence into the temporal encoder. However, this input does not contain the spectral information, so we also feed the spectrum data into this encoder and fuse these inputs with the encoded step $t$, i.e. $\boldsymbol{X}^{conv} = \boldsymbol{X}^{conv} + \boldsymbol{Z}^{conv} + t$. The same is true for the spectral encoder. The following is a detailed description of the temporal encoder and the spectral encoder.

**Temporal Encoder** The time sequence of IQ data is similar to traditional data like text. Therefore, we borrow the idea from Transformer, and apply the self-attention mechanism and residual paths throughout. Concretely, each block first expands the features through a feed-forward network (FFN) that injects non-linearity with the Gaussian-error linear unit (GELU) function, then lets a multi-head self-attention layer (MultiHead) aggregate long-range temporal dependencies. Formally,

$$\boldsymbol{X}^{out} = \boldsymbol{X}^{feat} + \text{MultiHead}(\boldsymbol{W}^Q \boldsymbol{X}^{feat}, \boldsymbol{W}^K \boldsymbol{X}^{feat}, \boldsymbol{W}^V \boldsymbol{X}^{feat}), \tag{9}$$

where $\boldsymbol{X}^{feat} = \boldsymbol{X}^{conv} + \text{FFN}(\boldsymbol{X}^{conv})$, and $\boldsymbol{W}^Q, \boldsymbol{W}^K, \boldsymbol{W}^V \in \mathbb{R}^{c \times c_h}$, $c_h = c/H$ denotes the dimension of each attention head, and $H$ is the number of head. Inspired by SpectrumFM, we next refine the feature map $\boldsymbol{X}^{out}$ and timestep embedding $t$ through a 1D point-wise convolution followed by a Gated Linear Unit (GLU). A 1D depth-wise convolution with kernel size 3 then harvests local signal structure, after which a second 1D convolution compresses the result into the final local representations.

$$\boldsymbol{X}^{local} = \boldsymbol{X}^{out} + \text{Conv1D}(\text{BN}(\text{Conv1D}(\text{GLU}(\text{Conv1D}(\boldsymbol{X}^{out}) + t), kernel = 3))), \tag{10}$$

where BN represents the BatchNorm. Finally, we use another FFN to combine the results above and output the temporal embeddings, i.e. $\boldsymbol{X}^{final} = \boldsymbol{X}^{local} + \text{FFN}(\boldsymbol{X}^{local})$.

**Spectral Encoder** The spectrum of IQ data typically has high amplitudes at certain frequencies, while other frequencies have very low amplitudes. This means that using the self-attention mechanism is not suitable for capturing sequence dependency information in the spectrum of IQ data. Therefore, we use the convolution layers to extract the local features and use the channel attention to obtain the key feature dimensions. Moreover, we also apply residual paths throughout the encoder. Specifically, we learn from the IQFormer and first encode the spectral dimension with a lightweight 1D depth-wise convolution followed by a 1D point-wise convolution, which is formulated as

$$\boldsymbol{Z}^{local} = \boldsymbol{Z}^{conv} + \text{Conv1D}(\text{Conv1D}(\text{BN}(\text{Conv1D}(\boldsymbol{Z}^{conv}, kernel = 3)))). \tag{11}$$

Next, we highlight the most informative channels through a channel attention module: the feature map is globally pooled, refined by a FFN network, and the resulting attention scores are multiplied with the original features. This re-weighting suppresses irrelevant bands and amplifies discriminative ones, yielding a more compact and representative spectral signature for downstream learning. We formulate the channel attention as

$$\boldsymbol{Z}^{feat} = \text{FFN}(\text{Pool}(\boldsymbol{Z}^{local})) * \boldsymbol{Z}^{local}. \tag{12}$$

Finally, we feed the feature map into the feedforward convolution network (FCN) that contains two 1D point-wise convolution layers for spectral embeddings, i.e. $\boldsymbol{Z}^{final} = \boldsymbol{Z}^{feat} + \text{FCN}(\boldsymbol{Z}^{feat})$.

During pre-training, the two embeddings are optimized by minimizing Equation 7. At the fine-tuning stage, we first apply global average pooling to each embedding, concatenate the resulting vectors into a single representation, and feed it to a classifier for the WSR task. The entire network is fine-tuned with standard cross-entropy loss.

## 4 EXPERIMENTS

In this section, we evaluate the performance of TS-DDAE on several datasets and WSR tasks. Specifically, we design a series of experiments to address the following questions. **Q1**: How effective is TS-DDAE in solving multiple WSR tasks? **Q2**: What is the quality of representation obtained by TS-DDAE pre-training? **Q3**: How do the temporal encoder and the spectral encoder contribute to the overall performance?

## 4.1 EXPERIMENTAL SETTINGS

The TS-DDAE is pre-trained and fine-tuned on NVIDIA Tesla A100 GPUs, using PyTorch Ansel et al. (2024) for implementation and Optuna Akiba et al. (2019) for hyperparameter optimization. The implementation details are listed in Appendix C. The hyperparameters used for TS-DDAE are listed in Appendix D. To evaluate the performance of TS-DDAE, we implement 11 models as our baseline models, which can be categorized as deep learning models, SOTA WSR models, and a WSR foundation model. Deep learning models contain ResNet Liu et al. (2017), MCNet Huynh-The et al. (2020), VGG O'Shea et al. (2018), CNN2 O'Shea et al. (2018), GRU2 Hong et al. (2017), CGDNN Njoku et al. (2021), Transformer Vaswani et al. (2017), MSNet Zhang et al. (2021). SOTA WSR models are AMC_Net Zhang et al. (2023) and IQFormer Shao et al. (2024). The WSR foundation model is SpectrumFM Zhou et al. (2025).

## 4.2 MAIN RESULTS (Q1)

In this part, we present the overall results of the TS-DDAE, which demonstrates the ability of our pre-trained TS-DDAE model to handle multiple WSR tasks. The Automatic Modulation Classification (AMC) and Wireless Technology Classification (WTC) results are presented in Table 1, where the model followed by the "probe" or "k-shot" is represented as the results of linear probe and few-shot learning described in Section 4.3. Moreover, we plot the performance across various SNRs of the AMC and WTC tasks in Figure 3. The analysis and results of the Anomaly Detection (AD) task on dataset ICARUS Roy et al. (2023) are presented in Appendix E.1.

Table 1: The overall results of different models (percentage), where the "Average" represents the average performance and the "Best" represents the best performance among all SNRs on 4 datasets (The best performance is represented in bold, and the second-best performance is underlined).

| Model | RML2016.10A | | RML2016.10B | | RML2018 | | TechRec | |
|---|---|---|---|---|---|---|---|---|
| | Average | Best | Average | Best | Average | Best | Average | Best |
| ResNet | 48.25 | 79.36 | 63.87 | 92.98 | 43.42 | 76.94 | 66.29 | 80.00 |
| MCNet | 60.37 | 90.86 | 57.18 | 87.43 | 52.70 | 89.22 | 67.12 | 85.68 |
| VGG | 39.99 | 62.18 | 38.11 | 56.63 | 42.53 | 66.73 | 79.83 | 94.07 |
| CNN2 | 54.18 | 80.68 | 54.70 | 79.24 | 41.67 | 61.16 | 60.52 | 71.38 |
| GRU2 | 60.39 | 90.86 | 63.93 | 93.34 | 52.57 | 82.44 | 37.80 | 39.09 |
| CGDNN | 51.62 | 76.59 | 47.90 | 70.68 | 35.34 | 52.56 | 49.27 | 58.15 |
| Transformer | 54.77 | 82.36 | 61.17 | 90.42 | 58.22 | 93.18 | 71.77 | 79.74 |
| MSNet | 58.94 | 88.59 | 63.37 | 93.34 | 57.73 | 91.92 | 85.80 | 97.71 |
| AMC_Net | 60.82 | 90.68 | 63.87 | 92.98 | 41.14 | 60.54 | 88.71 | 98.66 |
| SpectrumFM | 60.01 | 90.00 | 53.12 | 76.28 | 59.86 | 95.87 | 62.22 | 69.47 |
| IQFormer | **64.05** | 93.77 | 65.00 | 94.14 | 40.22 | 60.71 | 77.74 | 88.92 |
| TS-DDAE | 63.61 | **93.82** | **65.50** | **94.72** | **64.15** | **96.80** | **89.62** | **99.47** |
| TS-DDAE (probe) | 54.40 | 80.91 | 56.15 | 83.44 | 33.41 | 49.56 | 86.00 | 97.72 |
| TS-DDAE (25-shot) | 48.91 | 73.91 | 47.11 | 71.37 | 43.95 | 67.43 | 80.42 | 94.56 |
| TS-DDAE (100-shot) | 55.47 | 84.95 | 57.73 | 86.90 | 56.07 | 86.53 | 84.00 | 97.28 |

### 4.2.1 AUTOMATIC MODULATION CLASSIFICATION

**Task Description**. The AMC task is to automatically identify the modulation type used by the received signal under unknown prior parameters.

**Dataset**. We choose three most typical AMC benchmark datasets including RML2016.10A, RML2016.10B O'shea & West (2016), and RML2018 O'Shea et al. (2018) for this task. All datasets contain several modulation types under various Signal-to-Noise Ratio (SNR) conditions. The statistics of the datasets can be found in Appendix B, and we use 80% of the data for training, 20% of the data for evaluation.

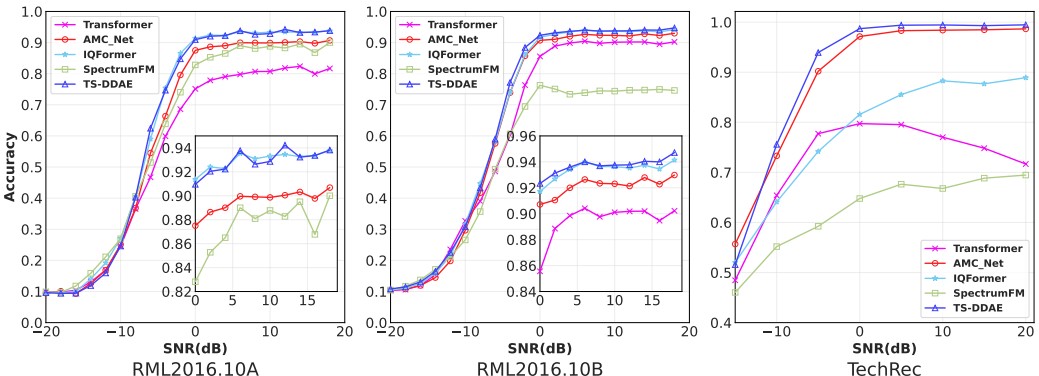

Figure 3: Visualization of performance under different SNR conditions (left: RML2016.10A, center: RML2016.10B, right: TechRec).

**Overall Performance**. The general AMC accuracy results are listed in the first three columns of Table 1, where we list the average and the best performance of different SNR conditions. From the results, we can find that compared with the deep learning baselines, the TS-DDAE outperforms the best baseline by 1.35% on average, indicating the effectiveness of our theory and model architecture. Compared with IQFormer, the SOTA method in AMC, the TS-DDAE also achieves SOTA performance in both "Average" and "Best" on RML2016.10B and RML2018 datasets. Though, the "Average" performance of TS-DDAE on RML2016.10A is slightly lower than that of IQ-Former, the "Best" performance is better. From this result, we believe that for simple datasets like RML2016.10A, TS-DDAE has not yet learned thoroughly, while for larger datasets like RML2018, TS-DDAE outperforms IQFormer by 23.07%, which means TS-DDAE has the potential to be trained on large-scale datasets and adapted to various AMC scenarios.

**Performance under various SNR conditions**. To better evaluate the robustness of TS-DDAE, we plot the performance of the models across various SNR conditions in Figure 3. We plot the results of dataset RML2016.10A, RML2016.10B, and choose the baseline models Transformer, AMC_Net, SpectrumFM, IQFormer for a clear view. At very low SNR (-20dB), the signal data is filled with noise, making it almost impossible to learn an appropriate signal representation. As the SNR increases, the model can more easily capture signal features, allowing the learned representation to be better applied to downstream classification tasks. Furthermore, our model performs well not only at high SNRs (greater than 0dB), but also at low SNRs (-10dB to 0dB). To have a more comprehensive view, we also provide the visualization of confusion matrix and t-SNE Maaten & Hinton (2008) of the model classification effect in the Appendix E.2.

### 4.2.2 WIRELESS TECHNOLOGY CLASSIFICATION

**Task Description**. The WTC task is used to determine to which communication system or protocol family the received wireless signal belongs without prior knowledge.

**Dataset**. We use the TechRec dataset for the WTC task. The TechRec contains Wi-Fi, LTE, and DVB-T three types of wireless technology. To reduce memory consumption, we slice each signal from the raw dataset into 1024-length segments and label each slice based on the label of the signal to which it belongs. Furthermore, to simulate real-world noise, we manually add Gaussian noise to generate different SNR signals ranging from -15dB to 20dB.

**Overall Performance**. The WTC accuracy results are listed in the fourth columns of Table 1, which demonstrate the superior performance the TS-DDAE compared to the other baseline models. For the "Best" performance among all the SNRs, the TS-DDAE can achieve a nearly 1.0 identification result. For the "Average" result, the TS-DDAE also achieves the SOTA performance, with about 11.88% improvement compared to the IQFormer, and 0.91% compared with the SOTA baseline.

**Performance under various SNR conditions**. The performance across all SNRs of TechRec is shown on the right of the Figure 3. Similar to the performance of the AMC task, the performance

improves as the SNR increases. In particular, starting at -10dB, our model consistently outperforms the SOTA AMC_Net model and significantly outperforms other baselines.

### 4.3 EVALUATION OF PRE-TRAINED MODEL CAPABILITIES (Q2)

To further evaluate the capabilities of our pre-trained model, we conduct the linear probe and few-shot experiments on the datasets RML2016.10A, RML2016.10B, RML2018, and TechRec.

**Linear Probe**. We freeze the pre-trained model parameters and only train the classifier with the training dataset. The results are listed in the row "TS-DDAE (probe)" of Table 1. Our linear probe results are comparable to the performance of some deep learning models. In particular, for the TechRec dataset, our results outperform all deep learning baseline models, which exhibits good linear separability of the pre-trained features.

**Few-shot Learning**. As the IQ signals contain $N$ classes under $M$ SNRs, we refer to the $N$-way $K$-shot sampling methods from CV, and define our $N$-way $K$-shot $M$-SNR sampling method, which is to select $N$ samples from each classification category and each SNR to form a dataset of $K \times N \times M$ IQ samples. In our work, we evaluate the 25-shot and 100-shot situation, whose results are presented in the rows "TS-DDAE (25-shot)" and "TS-DDAE (100-shot)" of Table 1, respectively. Even with fewer than 1% training data, the TS-DDAE still achieves an accuracy comparable to certain deep learning baselines. This result highlights the few-shot capabilities of our TS-DDAE model.

### 4.4 ABLATION STUDY (Q3)

Table 2: The ablation results (percentage) of TS-DDAE architecture.

| Model | RML2016.10A | | RML2016.10B | | TechRec | |
| --- | --- | --- | --- | --- | --- | --- |
| | Overall | Best | Overall | Best | Overall | Best |
| **TS-DDAE** | **63.61** | **93.82** | **65.50** | **94.72** | **89.62** | **99.47** |
| w/o temporal | 48.81 | 75.95 | 50.94 | 75.68 | 87.28 | 99.05 |
| w/o spectral | 62.97 | 93.09 | 64.79 | 93.73 | 37.48 | 47.58 |
| w/o interactive | 62.92 | 93.27 | 65.09 | 93.97 | 87.87 | 99.21 |
| w/ single noise | 63.01 | 92.59 | 65.12 | 94.14 | 89.27 | 99.21 |

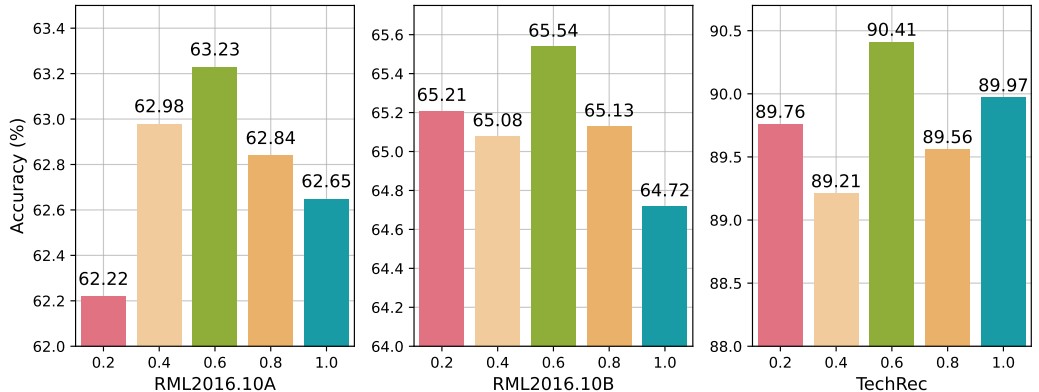

Figure 4: The accuracy of different $\lambda$ value ranging from 0.2 to 1.0 (left: RML2016.10A, center: RML2016.10B, right: TechRec).

In this part, we present the ablation study of TS-DDAE to evaluate the contribution of each component and the hyperparameter $\lambda$ analysis. The model parameter analysis is in Appendix E.3.

To validate the contribution, we evaluate four variants: (1) remove the temporal encoder (*w/o temporal*), (2) remove the spectral encoder (*w/o spectral*), (3) remove the interactive process and complete

denoising separately (*w/o interactive*), (4) use single noise as the objective (*w/ single noise*), and we keep the number of encoder layers consistent. The results are presented in Table 2. First, compared to the TS-DDAE, the ablation variants *w/o temporal* and *w/o spectral* exhibit performance degradation. For the datasets RML2016.10A and RML2016.10B of the AMC task, the temporal encoder plays a key role, with a decrease of about 15% in accuracy without the temporal encoder, while for the TechRec of the WTC task, the spectral encoder is more important, and the model without spectral encoder even fails to learn a proper model. This result demonstrates the contributions of each component in TS-Net. Then, we remove the interactive process proposed in the TS-Net, and complete denoising separately (*w/o interactive*). The performance degradation further demonstrates the necessity of our interactive design. To further illustrate the effectiveness of using two types of noise, we conduct the experiment using only one noise, retaining TS-Net, and transforming the optimization objective into the reconstruction of the original data (*w/ single noise*). The performance still degrades, which further illustrates that using different noise levels can optimize our training process. Fine-grained optimization helps the model learn the representation of the data, resulting in higher-quality features.

Then, we analyze the impact of hyperparameter $\lambda$ in Section 3.2, which is shown in Figure 4. As $\lambda$ gets closer to about 0.5, the model can achieve the best performance, which means that the two types of noise that we add will affect the model feature extraction to a certain extent. Keeping a balanced noise addition will improve the quality of the model.

## 5 CONCLUSION

In this work, we introduce TS-DDAE, a diffusion-style framework that pre-trains models by jointly noising and denoising the temporal and spectral perspectives of IQ data. The accompanying TS-Net uses self-attention for the temporal encoder and channel attention for the spectral encoder, letting each reinforce the other. Across several WSR benchmarks, TS-DDAE achieves SOTA performance, and ablation studies confirm that all component matters. We therefore expect TS-DDAE to serve as a solid starting point for future WSR foundation models.

### ACKNOWLEDGMENTS

This work is supported in part by the National Natural Science Foundation of China (No. 62550138, 62192784, 62572064, 62472329), BUPT Excellent Ph.D. Students Foundation (No. CX20241010), the Major Key Project of Peng Cheng Laboratory (PCL2024A08).

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

## A  THE DERIVATION OF THE BACKWARD PROCESS

We provide a detailed derivation of the Bayes's theorem, given Equation 4 and Equation 5,

$$p(x_{t-1}|x_t, x_0) = \frac{p(x_t|x_{t-1}) \cdot p(x_{t-1}|x_0)}{p(x_t|x_0)}$$

$$\propto \exp(-\frac{1}{2}[(\frac{(\beta_t^2 + \gamma_t^2)(\bar{\beta}_{t-1}^2 + \bar{\gamma}_{t-1}^2)}{1 - \bar{\alpha}_t^2})x_{t-1}^2 - 2(\frac{\alpha_t}{\beta_t^2 + \gamma_t^2}x_t + \frac{\bar{\alpha}_{t-1}}{\bar{\beta}_{t-1}^2 + \bar{\gamma}_{t-1}^2}x_0)x_{t-1} + C)],$$

(13)

where $C$ is a constant number that can be omitted.

## B  DETAILS OF THE DATASETS

We present the detailed statistics of the datasets that we used in Table 3, where SNR denotes the signal-to-noise ratio used for the datasets RML2016.10A, RML2016.10B, RML2018, and TechRec, and SIR denotes the signal-to-interference ratio used for the dataset ICARUS. For all datasets, we evenly sample 80% of the data for each category and SNR (SIR) for training and 20% for testing.

Table 3: The statistics of the datasets used in this work.

| Dataset | RML2016.10A | RML2016.10B | RML2018 | TechRec | ICARUS |
|---|---|---|---|---|---|
| number of samples | 220,000 | 1,200,000 | 2,555,904 | 202,762 | 673,200 |
| length of each sample | 128 | 128 | 1,024 | 1,024 | 1,024 |
| number of classes | 11 | 10 | 24 | 3 | 2 |
| min SNR (SIR) | -20 | -20 | -20 | -20 | 0 |
| max SNR (SIR) | 18 | 18 | 30 | 20 | 10 |
| SNR (SIR) interval | 2 | 2 | 2 | 5 | 5 |

Table 4: The pre-training time of the TS-DDAE.

| | RML2016.10A | RML2016.10B | RML2018 | TechRec |
|---|---|---|---|---|
| Time(s) | 1,791 | 1,166 | 1,178 | 1,791 |

## C  IMPLEMENTATION DETAILS

Our pre-training initialization method is the default PyTorch initialization method, i.e., a uniform distribution bounded by 1/sqrt(in_features). Our pre-training learning rate strategy is the "ReduceLROnPlateau", with parameters: mode="min", factor=0.5, patience=250, min_lr=1e-12. To ensure a fairer comparison with baselines, we pretrain our model on each dataset separately. Furthermore, to save time, we implement an early stop mechanism with a patience of 1000. The overall pre-training time is shown in the Table 4.

## D Hyperparameter usage in experiments

We present all hyperparameters used in our experiments in Table 5, where we list the description and the value of hyperparameters. Furthermore, we use the same hyperparameters for all of our datasets.

Table 5: The hyperparameters used in our experiments.

| Parameter name | Description | Value |
|---|---|---|
| num_layers | The number of TS-Net layers | 4 |
| max_step | The maximal diffusion steps | 3000 |
| timestep | The step used in fine-tuning | 4 |
| ratio | The ratio between the temporal noise and the spectral noise | 0.414 |
| min_noise | The minimal noise added to signals | 5.45e-06 |
| max_noise | The maximal noise added to signals | 0.0072 |

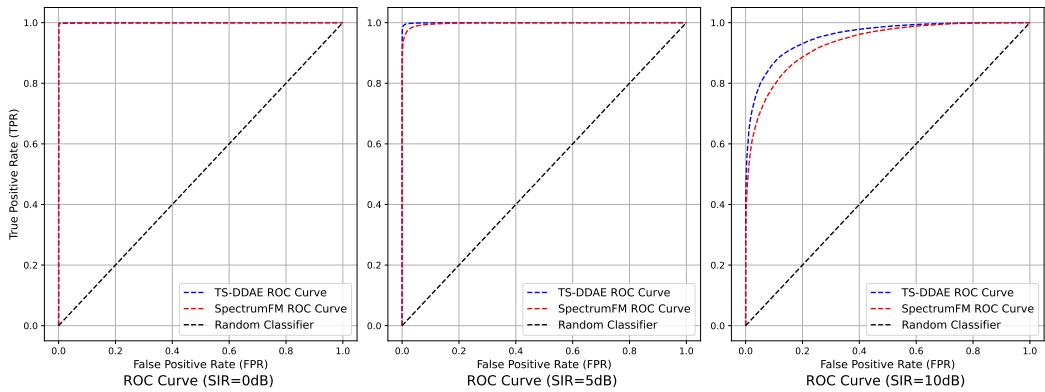

Figure 5: AUC_ROC curve of the anomaly detection under different SIR (left: 0dB, center 5dB right: 10dB).

## E Additional Experiments

### E.1 Anomaly Detection

In this section, we supplement the results of the anomaly detection (AD) task.

**Task Description**. The AD task aims to detect whether other unknown signals are mixed in the received signal.

**Dataset**. We use a part of the ICARUS dataset for the AD task. The AD task of the ICARUS dataset is to detect whether the normal LTE signal is mixed with the anomalous DSSS signal. Similarly to the WTC settings, we also slice each signal into 1024-length segments. The signal-to-interference ratio (SIR) varies from 0dB to 10dB, which is extracted from the dataset metadata.

**Overall Performance**. The results are presented in Figure 5. When SIR is 0dB or 5dB, the TS-DDAE can get an AUC nearly to 1.0, which demonstrates excellent anomaly signal detection capabilities. As the SIR increases to 10dB, the power of the DSSS signals also increases, the detection performance drops, while still maintaining a high performance (over 0.9). Compared with SpectrumFM, the TS-DDAE can still achieve a higher performance, which further indicates the effectiveness of the designed model.

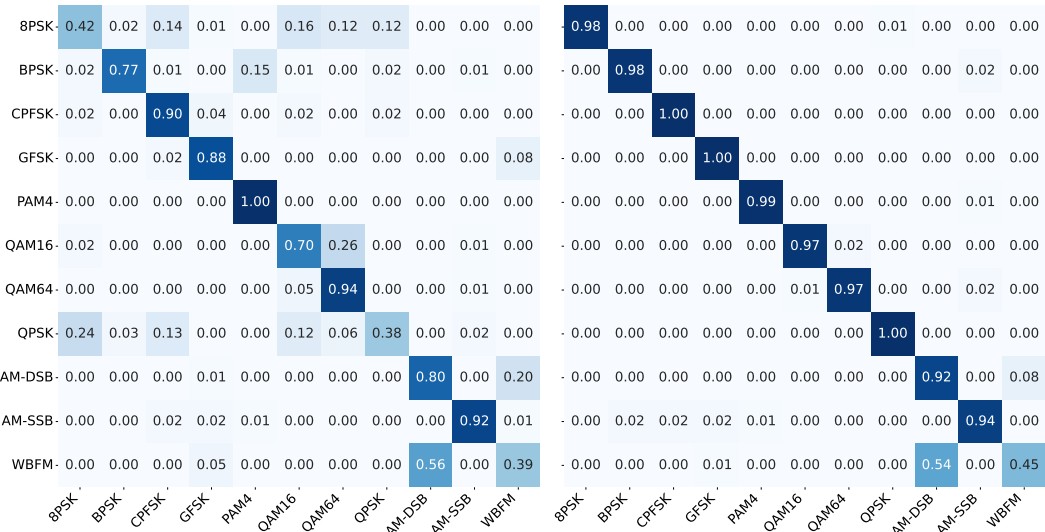

Figure 6: Normalized confusion matrix on RML2016.10A dataset (left: -4dB, right: 2dB).

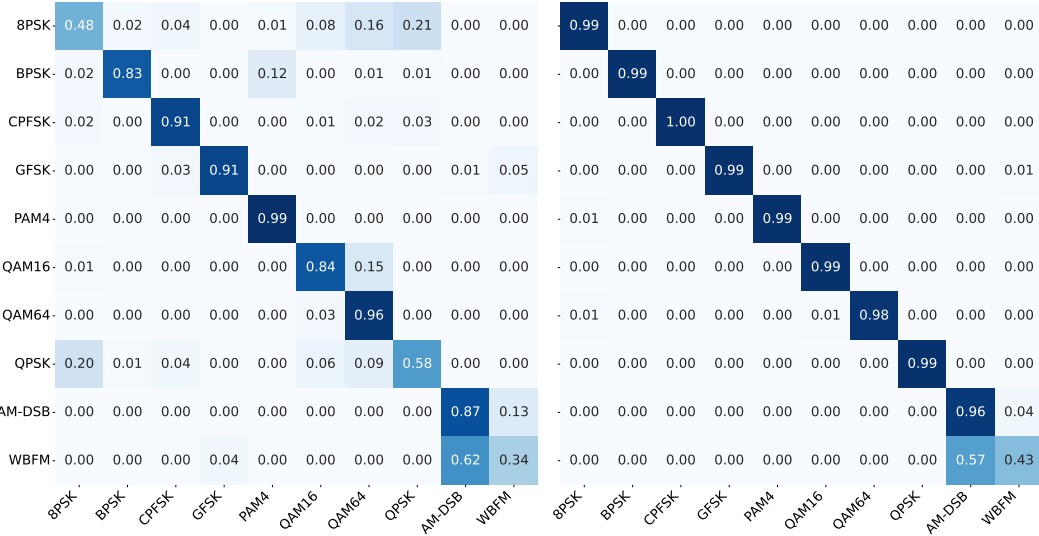

Figure 7: Normalized confusion matrix on RML2016.10B dataset (left: -4dB, right: 2dB).

## E.2 VISUALIZATION OF THE TS-DDAE OUTPUT

In this section, we visualize the output of the TS-DDAE model in the AMC task. The results of the figures are consistent with the conclusions of the SpectrumFM and the IQFormer, and can further verify the the correctness and effectiveness of the TS-DDAE.

First, we give the normalized confusion matrix of TS-DDAE results on the RML2016.10A and the RML2016.10B dataset with -4dB and 2dB SNR, which is shown in Figure 6 and Figure 7. From the results, we can find that under low SNR conditions, noise will seriously affect the TS-DDAE's distinction between the three modulation types: 8PSK, QPSK and WBFM. When the SNR reaches 2dB, the model can already distinguish most modulation types well, even distinguishing some modulation types with 100% accuracy. However, the model can hardly distinguish the WBFM modulation type from the AM-DSB, which is consistent with the conclusions of the SpectrumFM and the IQFormer.

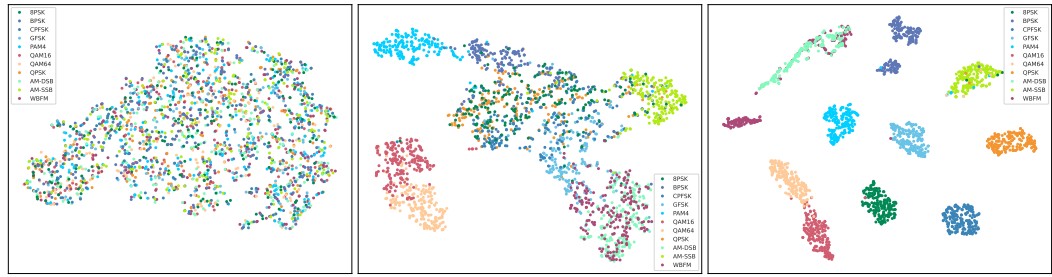

Figure 8: Visualization of feature output with t-SNE on RML2016.10A dataset (left: -20dB, center: -6dB, right: 12dB).

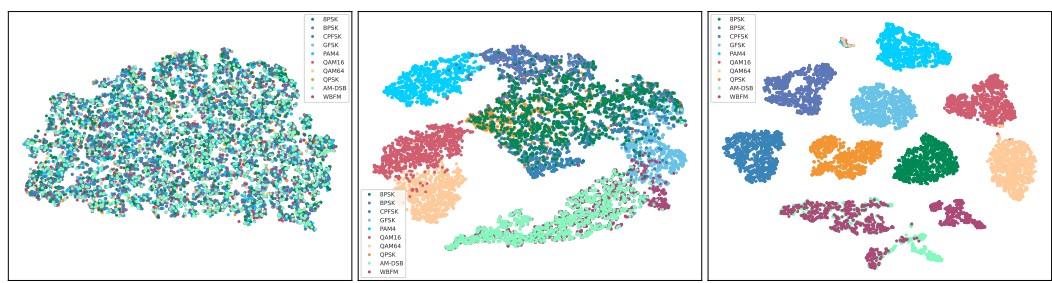

Figure 9: Visualization of feature output with t-SNE on RML2016.10B dataset (left: -20dB, center: -6dB, right: 12dB).

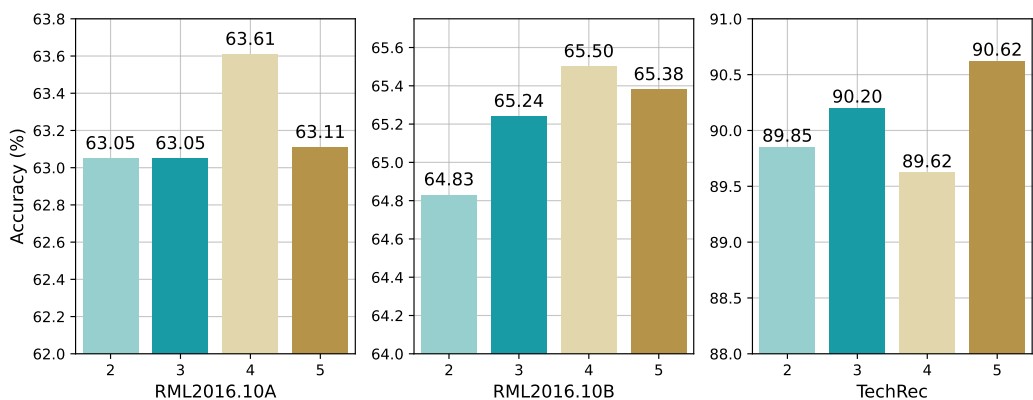

Figure 10: The accuracy of different number of layers ranging from 2 to 5 (left: RML2016.10A, center: RML2016.10B, right: TechRec).

Then, we present the t-SNE visualizations of the RML2016.10A and the RML2016.10B dataset under -20dB, -6dB, and 12dB for TS-DDAE. The results are shown in Figure 8 and Figure 9. At a -20dB SNR, the data is filled with noise, leading the model to classify data randomly. Consequently, the various modulation categories appear mixed together in the t-SNE figure. At a low SNR (-6dB), our TS-DDAE already demonstrates a certain degree of discrimination, with many modulation categories clearly distinguished. At a high SNR (12dB), the model's discrimination is even better. Furthermore, at 12dB, the model fails to distinguish WSFM and AM-DSB modulation categories particularly well, which is consistent with the conclusions in Figure 6 and Figure 7.

### E.3 THE EFFECT OF THE MODEL SIZE

In this section, we provide experiments of different numbers of encoder layers in TS-Net. Each additional layer means adding one temporal encoder and one spectral encoder simultaneously. The results are shown in Figure 10. As the number of layers drops, the performance of RML2016.10A and RML2016.10B also drops. The proper number of layers is 4, which is used in our main experiments. For dataset TechRec, the performance reaches the highest when the number of layers is 5, which we think that the scale of the data is relatively large, and a large model can get better signal representations.

## F   THE USE OF LARGE LANGUAGE MODELS (LLMs)

We use the LLM Kimi K2 to aid or polish writing of this paper.

