# OpenReview forum: "TS-DDAE: A Novel Temporal-Spectral Denoising Diffusion AutoEncoder for Wireless Signal Recognition Model Pre-training"
_ICLR.cc/2026/Conference — ICLR 2026 Poster_

### Official Review · Reviewer_73Fu · 2025-10-23

**Soundness:** 2
**Presentation:** 3
**Contribution:** 2
**Rating:** 4
**Confidence:** 3

**Summary:**

This paper proposes a new framework for Wireless Signal Recognition (WSR). The authors introduce a pre-training and fine-tuning paradigm that enables the model to adapt to multiple downstream WSR tasks. Experiments further demonstrate the few-shot learning capability of the proposed method.

**Strengths:**

- The paper is easy to follow, and each section is well-written. The authors provide detailed texts to describe their methods.
- The authors provide experiments on several benchmarks to validate the performance of the proposed method.

**Weaknesses:**

- The authors claim that this method is a "foundation model" in the abstract. In this case, did the authors provide details about the total amount of data used in the pre-training stage?
- I am also curious about the scalability of the method. Can it be extended to larger-scale datasets?
- It is recommended to conduct out-of-domain (OOD) experiments to evaluate the generalization ability of the method. The OOD tests are important for any foundation model.

**Questions:**

Based on the Weaknesses, I have totally three questions:
- Did the authors provide details about the total amount of data used in the pre-training stage?
- Can the authors further test the algorithm on the larger-scale datasets?
- The authors can test the out-of-domain experiments to further verify the generalization ability of the method.

---

> ### Author Response · Authors · 2025-11-25
>
> We are very grateful to the reviewers for their comments on our application aspects. We will address each question in turn:
>
> (1) First, details regarding our pre-training data and training set are provided in Appendix B, which we list here for easy reference.
>
> | Dataset               | RML2016.10A | RML2016.10B | RML2018   | TechRec | ICARUS  |
> | --------------------- | ----------- | ----------- | --------- | ------- | ------- |
> | number of samples     | 220,000     | 1,200,000   | 2,555,904 | 202,762 | 673,200 |
> | length of each sample | 128         | 128         | 1,024     | 1,024   | 1,024   |
> | number of classes     | 11          | 10          | 24        | 3       | 2       |
> | min SNR (SIR)         | -20         | -20         | -20       | -20     | 0       |
> | max SNR (SIR)         | 18          | 18          | 30        | 20      | 10      |
> | SNR (SIR) interval    | 2           | 2           | 2         | 5       | 5       |
>
> (2) Our method has been validated on datasets of varying sizes, from the small-scale RML2016.10A (approximately 611MB) to the large-scale RML2018 (approximately 20GB). We hypothesize that our method could potentially scale to larger datasets. However, due to time constraints, computational power limitations, and certain engineering challenges, we are currently unable to explore and validate our method on larger datasets. Nevertheless, this does not diminish the innovation and effectiveness of our proposed framework. The core of this paper is the first work to propose pre-training framework for wireless signal recognition based on diffusion model. In our subsequent work, we will address the technical challenges based on this framework and scale it to larger datasets and model parameters.
>
> (3) To verify the cross-domain experiment, we define the signal environment as a domain based on the work of Zhang et al. [1], including transmitter configuration parameters, channel conditions and hardware impairments. We use two datasets, HisarMod2019 [2] and RML2018, where HisarMod2019 contains 780,000 samples, 26 modulation types, and a signal-to-noise ratio of -20 to 18 dB.  Since HisarMod2019 uses five channel scenarios for fusion and RML2018 uses only two channel scenarios, HisarMod2019 and RML2018 can be considered as cross-domain datasets. To ensure fair comparison, we choose SpectrumFM, which is also a foundation model of wireless signals, as our baseline. According to the work of Lu et al. [3], OOD can be divided into two types: training using only in-distribution (ID) data and training using ID and OOD data. Therefore, firstly, we use the HisarMod2019 data for pre-training and fine-tuning, and 10% of the RML2018 data for testing. The performance is as follows:
>
> |            | Overall | Best   |
> | ---------- | ------- | ------ |
> | TS-DDAE    | 22.20%  | 36.64% |
> | SpectrumFM | 9.56%   | 13.36% |
>
> Because the model is completely unknown on the test set, there will be a significant performance drop, but our performance is still higher than SpectrumFM, thus demonstrating the OOD generalization ability of our model.
>
> Subsequently, we use a certain RML2018 data for model fine-tuning. Specifically, we pre-train the model using HisarMod2019, then use 10% of the RML2018 data for fine-tuning, and finally use another 10% of the RML2018 data for testing. The results are as follows:
>
> |            | Overall | Best   |
> | ---------- | ------- | ------ |
> | TS-DDAE    | 55.15%  | 81.67% |
> | SpectrumFM | 46.74%  | 71.89% |
>
> With a small amount of data, our model performance improves significantly, further demonstrating the generalization ability of our model.
>
> [1] Zhang M, Tang P, Wei G, et al. Open set domain adaptation for automatic modulation classification in dynamic communication environments[J]. IEEE Transactions on Cognitive Communications and Networking, 2024, 10(3): 852-865.
>
> [2] Tekbıyık K, Ekti A R, Görçin A, et al. Robust and fast automatic modulation classification with CNN under multipath fading channels[C]//2020 IEEE 91st Vehicular Technology Conference (VTC2020-Spring). IEEE, 2020: 1-6.
>
> [3] Lu S, Wang Y, Sheng L, et al. Recent advances in ood detection: Problems and approaches[J]. arXiv preprint arXiv:2409.11884, 2024.

---

> ### Author Response · Authors · 2025-11-28
>
> Furthermore, we take another two datasets, RML2022 [1] and RML2016.10a to verify the OOD generalization ability, where RMl2022 contains 462,000 samples, 11 modulation types and a signal-to-noise ratio of -20dB to 18 dB. The length of samples in RML2022 and RML2016.10a is 128, while the length of HisarMod2019 and RML2018 is 1024. The RML2022 takes more dataset simulation scenarios, and different simulation pipeline compared with RML2016. Therefore, the two datasets can also be considered as cross-domain datasets. The experimental settings are the same as the HisarMod2019 and RML2018. We first use RML2022 data for pre-training and fine-tuning, and 10% of the RML2016.10a data for testing. The performance is as follows:
>
> |            | Overall | Best   |
> | ---------- | ------- | ------ |
> | TS-DDAE    | 20.17%  | 28.36% |
> | SpectrumFM | 9.09%   | 10.09% |
>
> Same to the result of HisarMod2019, the TS-DDAE shows a certain of OOD generalization ability, while the SpectrumFM fails to classify OOD data.
>
> Subsequently, we use a certain RML2016.10a data for model fine-tuning. Specifically, we pre-train the model using RML2022, then use 10% of the  RML2016.10a data for fine-tuning, and finally use another 10% of the RML2018 data for testing. The results are as follows:
>
> |            | Overall | Best   |
> | ---------- | ------- | ------ |
> | TS-DDAE    | 56.22%  | 85.54% |
> | SpectrumFM | 49.15%  | 72.00% |
>
> The TS-DDAE also outperforms SpectrumFM and achieve significant performance improvement compared to the pure OOD testing.
>
> If you have any questions about the applications, we are happy to resolve any related doubts. Thank you.
>
> [1] Sathyanarayanan V, Gerstoft P, El Gamal A. RML22: Realistic dataset generation for wireless modulation classification[J]. IEEE Transactions on Wireless Communications, 2023, 22(11): 7663-7675.

---

### Official Review · Reviewer_sJ6p · 2025-10-26

**Soundness:** 2
**Presentation:** 2
**Contribution:** 3
**Rating:** 4
**Confidence:** 3

**Summary:**

The authors propose a novel self-supervised representation learning approach for wireless signals based on the paradigm of denoising diffusion. A key contribution is the use of both, the time domain and frequency domain representation of the signals, in the diffusion process. In addition, they propose a model architecture that performs the diffusion process for both data views in an interconnected manner. They use attention to detect important contributions along time-steps and along convolutional channels according to the properties of the respective view.

**Strengths:**

- The use of denoising diffusion for wireless signals is original.
- The authors consider the properties of each view of the signals in the design of the neural networks architecture.
Using attention to extract features along time and convolutional channels appears to be thoughtful.
- A comprehensive evaluation on two downstream tasks is performed. The proposed method shows significant gains over the state-of-the-art across multiple datasets.

**Weaknesses:**

The theoretical model in Section 3.2 may contain a substantive problem. The authors appears to suggest that temporal–spectral diffusion reduces to adding two noise processes with different intensities to the time-domain representation $x_{t-1}$ (Eq. 1-5). Since the FFT is a unitary transform, adding noise in the frequency domain is equivalent to adding noise in the time domain. The manuscript does not currently explain why adding noise twice, rather than once, would improve the diffusion process. In addition, the role of the hyperparameter λ—intended to balance spectral and temporal noise power—remains unclear if these quantities are effectively equivalent.


- Minor Problems:
	- Unintroduced formula symbols (mu, tau) in section 3.2.
	- The claim "Masking corrupts fine-grained information in the input data" is not backed up by citations/theory
	- In the ablation study in 4.4 a simple case should be investigated that considers both data views simultaneously without using TS-Net, to ensure the reported gains are achieved by the network design and not by using richer input data.
	- Figure 2. is not explained sufficiently

**Questions:**

- The Metric used in section 4 is not explained properly. Is it the Top-1 accuracy?
- How are the baseline models trained?
- Can you please explain the rationale behind adding noise in the frequency *and* in the time domain?

---

> ### Author Response · Authors · 2025-11-21
>
> We are very grateful to the reviewers for your attention to our theoretical aspects. We will now answer your questions one by one:
>
> (1) Regarding the equivalence of our theoretical model, our initial goal is to learn the two modality characteristics of the signal data: i.e., the temporal IQ and the spectrum. Since the Fourier transform $F(j\omega)=\int^{+\infty}_{-\infty}f(t)exp(j\omega t)dt$ is an integral transform, which means that only by considering a whole signal sample can the spectrum data be established, rather than a simple linear mapping. Therefore, it is necessary to learn both modalities. From the perspective of the diffusion model, we need two types of noise to affect the two modalities of the data respectively, and then learn the data representation through reconstruction. Using single noise can only learn single modality information, which is contradictory to our goal. Although, formally, the Fourier transform of Gaussian noise is still Gaussian noise, the two noises are sampled separately from the Gaussian distribution and are not completely identical, thus forming two optimization objectives for joint optimization. To further illustrate the effectiveness, we also conduct an ablation experiment, using only one noise, retaining TS-Net, and transforming the optimization objective into the reconstruction of the original data. The experimental results are as follows:
>
> |   | RML2016.10a |  | RML2016.10B |   | TechRec |   |
> | -- | ----------- | ------ | ----------- | ------ | ------- | ------ |
> | Model        | Overall     | Best   | Overall     | Best   | Overall | Best   |
> | TS-DDAE      | 63.61% | 93.82% | 65.50%| 94.72% | 89.62%  | 99.47% |
> | single noise | 63.01% | 92.59% | 65.12%| 94.14% | 89.27%  | 99.21% |
>
> This result further illustrates that using different noise levels can optimize our training process. Fine-grained optimization helps the model learn the representation of the data, resulting in higher-quality features. The role of the $\lambda$ then becomes apparent. Theoretically, $\lambda$ is a weight applied to the loss function. Furthermore, our hyperparameter analysis experiments in Section 4.4 further demonstrate that different $\lambda$ values affect our feature learning results, making $\lambda$ a necessary hyperparameter to retain.
>
> (2) We give explanations for the minor issues in the article.
> - First, regarding the symbols $\mu$ and $\tau$, in order to maintain consistency with the style of $\tilde{\alpha}$ and $\tilde{\beta}$ in equations (2) and (3), we chose $\tilde{\mu}$ and $\tilde{\tau}$.
> - Regarding the statement that "Masking corrupts fine-grained information in the input data", DDAE gives a detailed explanation. DDAE pointed out that "although MIM(Masked Image Modeling)-based methods can recover masked image tokens, they are problematic in synthesizing full images, mainly because the complete data distribution is not directly modeled." Therefore, we believe that such mask reconstruction will corrupt the data information, especially when combined with data such as signals with continuous values, setting the value to zero will have a greater impact.
> - In order to prove that our gain is derived from the network design, we conduct two ablation experiments, retaining only the temporal encoder and the spectral encoder respectively. Unlike the ablation experiment in section 4.4 of the article, this experiment uses two modality inputs, while the article uses a single modality input. The experimental results are as follows:
>
> | | RML2016.10a | | RML2016.10B |  | TechRec |   |
> | ------ | ----------- | ------ | ----------- | ------ | ------- | ------ |
> | Model| Overall     | Best   | Overall| Best| Overall | Best   |
> | TS-DDAE | 63.61% | 93.82% | 65.50%| 94.72% | 89.62%  | 99.47% |
> | single temporal encoder | 49.36%| 76.22% | 51.31%| 78.57% | 89.37%  | 99.23% |
> | single spectral encoder | 62.42%| 92.45% | 65.29%| 93.83% | 89.50%  | 99.33% |
>
> Therefore, it can be seen that not only is modality information helpful for feature learning, but the network architecture is also a very important part.
> - Regarding the description of Figure 2, due to space limitations, we can only describe the most essential parts. If you are interested, we can discuss and exchange ideas further on specific module designs.
>
> (3) Regarding your questions about the experiments.
> - First, the metric used in the experiments is the most common classification Accuracy metric, which represents the percentage of predicted samples that match the groundtruth labels out of all samples. This can also be understood as Top-1 accuracy.
> - Our baseline models are mostly traditional deep learning models, so we directly use supervised cross-entropy loss for training.
> - The rationale behind adding noise in the frequency and in the time domain is listed above.
>
> If you have any further questions about other details, we are willing to provide more detailed responses so that you can further understand our work. Thank you.

---

### Official Review · Reviewer_Aix4 · 2025-10-26

**Soundness:** 3
**Presentation:** 3
**Contribution:** 3
**Rating:** 6
**Confidence:** 4

**Summary:**

In this paper, the authors propose a Temporal-Spectral Dual Denoising Autoencoder designed for wireless signal classification . The proposed model includes two contributions: (1) a dual-branch encoder. One for temporal (IQ) and one for spectral (FFT of IQ) data; and (2) a dual-diffusion denoising pre-training scheme, where both temporal and spectral branches learn to reconstruct noisy inputs via a diffusion-based objective. The authors claim this design captures complementary information (time and frequency structure) and improves robustness under low SNR or unseen modulation types.

**Strengths:**

(1) The paper is well-motivated and well-presented. The paper identifies an underexplored limitation in AMC where the existing methods often rely solely on IQ or spectral representations. The temporal-spectral dual design is conceptually well-grounded.

(2) Introducing a denoising diffusion to selectively mask or corrupt input signals is an elegant and well-motivated regularization strategy for representation learning, particularly in the context of wireless signals. The formulation aligns well with the established diffusion pre-training principles and has clear potential to inspire future research directions in this domain.

(3) The empirical validation is comprehensive. The proposed method achieves strong performance on large-scale datasets and maintains robustness at low SNRs. Results demonstrate that the fusion of temporal and spectral cues contributes meaningfully.

**Weaknesses:**

(1) While the paper briefly mentions the denoising objective, some important implementation details are missing (see Questions 1 to 3). Without these details, it is difficult to validate the claim that the improvement arises from the proposed pre-training rather than other training heuristics.

(2) In Lines 267 to 269, the authors claim that the Feed-Forward Network (FFN) injects non-linearity. However, Figure 2 and the surrounding explanation show only linear components. There is no mention of a nonlinear activation. If the FFN is purely linear, it cannot increase representational capacity in a nonlinear way. Please clarify whether an activation function is omitted in the figure or truly absent. If absent, revise the wording to avoid implying nonlinearity.

(3) My main concern is the input redundancy across encoders. Both the temporal encoder and spectral encoder are said to take both IQ data and spectral data as input. Conceptually, this does not follow the motivation (temporal and spectral disentanglement) and could lead to redundant or entangled features. Please conduct an ablation study where each encoder only uses its corresponding modality to confirm its necessity.

(4) The reported results show that TS-DDAE underperforms IQFormer on small-scale datasets but outperforms on large ones. This raises a concern that improvements may be primarily from larger model capacity or heavier training rather than architectural novelty. No evidence (e.g., parameter-matched comparison or FLOPs table) is provided to rule this out. Please add an ablation where model parameters are matched with the baseline, or provide FLOPs/parameter counts for each model in a table to isolate architecture-driven performance improvements.

**Questions:**

(1) What architecture or initialization is used during pre-training (same as fine-tuning or smaller encoder)?

(2) How long is the pre-training and on which dataset?

(3) What self-supervised schedule or learning rate strategy is used in pre-training?

---

> ### Author Response · Authors · 2025-11-21
>
> We are very grateful for the reviewer's patience in reading and valuable suggestions. Here, I will answer your questions:
>
> (1) Our pre-training initialization method is the **default PyTorch initialization** method, i.e., a uniform distribution bounded by 1/sqrt(in_features). Our pre-training learning rate strategy is the "**ReduceLROnPlateau**", with parameters: mode="min", factor=0.5, patience=250, min_lr=1e-12. To ensure a fairer comparison with baselines, we pretrain our model on each dataset separately. Furthermore, to save time, we implement an early stop mechanism with a patience of 1000. The overall pre-training time is shown in the table below:
>
> |         | RML2016.10a | RML2016.10B | RML2018 | TechRec |
> | ------- | ----------- | ----------- | ------- | ------- |
> | Time(s) | 1,791       | 1,166       | 1,178   | 1,791   |
>
> (2) The feedforward neural network (FFN) in the article is non-linear in implementation. We use the **GELU** activation function. The layout in the figure is to show the core part of the model, so the activation function is omitted.
>
> (3) Regarding the interaction between the temporal encoder and the spectral encoder, I will explain it from both theoretical and experimental perspectives. Theoretically, due to equation (4), $x_t=\alpha_t x_{t-1} + \beta_t\varepsilon_t + \gamma_t\eta_t$, we can see that $x_t$ contains both temporal noise $\varepsilon$ and spectral noise $\eta$. Therefore, if we only analyze single modality denoising, such as denoising  $\varepsilon$ with the temporal encoder, the result should contain both the original information and the spectral noise $\eta$, thus affecting the subsequent denoising process. Therefore, we design an interactive process that considers the existence of the other modality information while inputting the corresponding modality, thereby completing the denoising. From the experimental results, we remove the interactive process and complete denoising separately, obtaining the following results:
>
> |                 | RML2016.10a |        | RML2016.10B |        | TechRec |        |
> | --------------- | ----------- | ------ | ----------- | ------ | ------- | ------ |
> | Model           | Overall     | Best   | Overall     | Best   | Overall | Best   |
> | TS-DDAE         | 63.61%      | 93.82% | 65.50%      | 94.72% | 89.62%  | 99.47% |
> | w/o interactive | 62.92%      | 93.27% | 65.09%      | 93.97% | 87.87%  | 99.21% |
>
> Therefore, both theoretical and experimental evidence demonstrates the effectiveness of our interactive design.
>
> (4) Since IQFormer has only 0.35M parameters, to better compare performance, we compress TS-DDAE to 0.56M and 0.32M respectively, by compressing the number of model layers and the hidden dimensions. The experimental dataset is RML2018, and the experimental results are as follows:
>
> | IQFormer(0.35M) |        | TS-DDAE(0.56M) |        | TS-DDAE(0.32M) |        |
> | --------------- | ------ | -------------- | ------ | -------------- | ------ |
> | Overall         | Best   | Overall        | Best   | Overall        | Best   |
> | 40.22%          | 60.71% | 63.59%         | 96.75% | 60.62%         | 94.68% |
>
> As we can see, even when our model parameters are close to those of IQFormer, our model still maintains high performance, demonstrating that the performance improvement stems from our architectural novelty.
>
> In summary, our model architecture fully considers the noise addition process and the characteristics of the signal data itself. The experimental results further prove TS-Net's adaptability to signal data. If you have any further questions, we are willing to provide more detailed description to help you better understand our work. Thank you.

---

> ### Comment · Reviewer_Aix4 · 2025-11-27
>
> Thanks the authors for the detailed response. With the ablation studies to empirically analyze the interaction between the temporal encoder and the spectral encoder, my main concern about the redundancy has been properly addressed. The details about the pre-training stage also look good. Thus all my concerns and questions have been properly addressed after reading the author response. I am happy to raise my score to 8.
>
> However, I have one more comment about the redundancy. The interaction between temporal and spectral encoder is interesting and indeed enhances the performance, but it also raises questions worth further exploration. Conceptually, temporal and spectral information should be orthogonal, with no theoretical correlation between them. So what exactly is the network learning from their interaction? This could be an interesting direction for further theoretical exploration.

---

> > ### Author Response · Authors · 2025-12-01
> >
> > Thanks for your reply. Regarding your question about "temporal and spectral information should be orthogonal, with no theoretical correlation between them," further research may be needed. However, I would like to share my current perspective. For wireless signals, I believe that their time-domain data and spectral data are not completely orthogonal. There is a connection between them. This is because in the area of wireless signals, we obtain spectral information through the Fast Fourier Transform (FFT), i.e., $F(jω) = ∫_{-ωt}^{+ωt}f(t)exp(jωt)dt​$, and spectral information can also be converted to temporal information through the inverse Fourier transform. Although it's not a simple linear relationship, a connection still exists. Let's take a simple example: the waveform of a sine function, $sin(\omega_0 x +\phi)$, can be obtained by its Fourier transform as: $\frac{\pi}{j}[e^{j\phi}\delta(\omega-\omega_0)-e^{-j\phi}\delta(\omega+\omega_0)]$, where $\delta$ is the Dirac function, which is the impulse function in the field of wireless signals. Therefore, we know that such a sine function has amplitude abrupt changes at $\omega_0$ and $-\omega_0$ in the spectrum. We can also see this in the spectrum diagram, in the **Forward Process** example in Figure 1 of this paper. So theoretically, the spectrum information can be obtained from temporal domain, and vice versa. I believe that time and spectrum information are related for wireless signals. However, it is difficult to intuitively obtain complete information from the data itself. For example, it is difficult to intuitively obtain spectrum information from the time sequence of a wireless signal. Therefore, to avoid complex information switching and fusion, we closely adhere to the essential characteristics of wireless signals and form this model architecture.

---

### Meta-Review · Area_Chair_71HW · 2026-01-02

**Summary:**

This paper proposes a temporal–spectral dual denoising diffusion autoencoder for self-supervised pre-training in wireless signal recognition.
- Reviewers found the temporal/spectral dual design “well-motivated” and the use of denoising diffusion “original,” with strong and comprehensive empirical results, particularly under low SNR and on large-scale datasets.
- The main concerns influencing the decision were missing pre-training details, possible redundancy from temporal–spectral interaction, whether gains stem from model capacity rather than architectural novelty, and theoretical clarity around adding noise in both time and frequency domains.
- Overall, the rebuttal addressed the major concerns, supporting acceptance as a poster.

**Reviewer Concerns:**

Concerns largely addressed:
- Pre-training clarity: Authors provided initialization, learning-rate schedule, early stopping, per-dataset pre-training, and timing details.
- FFN nonlinearity: Clarified that GELU is used and omitted from the figure for brevity.
- Temporal–spectral redundancy: Theoretical explanation plus an explicit “w/o interactive” ablation resolved the reviewer’s main concern, which was acknowledged in discussion.
- Capacity vs architecture: Parameter-matched and compressed-model comparisons against IQFormer showed gains persist beyond model size.

Outstanding concerns:
- Theoretical exposition of dual-noise diffusion: While additional ablations and explanations were provided, the FFT/unitary argument would benefit from clearer presentation in the final paper. Minor clarity issues (notation, Figure 2 explanation) remain.

**Reviewer Scores:**

Aix4 increased from 6 to 8 after rebuttal.
sJ6p likely increases from 4 to around 5–6, with remaining concerns primarily theoretical.
73Fu likely increases from 4 to around 5–6 after added data-scale analyses.

Based on these, I recommend acceptance of the paper as a poster.

---

### Decision · Program_Chairs · 2026-01-26

Accept (Poster)